# Advanced Customer Behavior Tracking and Heatmap Analysis with YOLOv5 and DeepSORT in Retail Environment

Mohamed Shili [1,*], Sudarsan Jayasingh [2,*] and Salah Hammedi [3,4]

1 Innov'COM Laboratory, National Engineering School of Carthage, University of Carthage, Charguia II, Carthage 2035, Tunisia
2 Department of Management Studies, Sri Sivasubramaniya Nadar College of Engineering, Kalavakkam 630110, Tamil Nadu, India
3 Networked Objects, Control, and Communication Systems (NOCCS) Laboratory, ENISo, University of Sousse, Sousse 4054, Tunisia; salah.hammedi@enim.u-monastir.tn
4 Electrical Engineering Department, National School of Engineers of Monastir, Monastir 5000, Tunisia
* Correspondence: mohamed.shili@fst.utm.tn (M.S.); sudarsanj@ssn.edu.in (S.J.)

**Abstract:** This paper presents a computer-vision-based approach designed to enhance product placement and sales strategies in physical retail stores through real-time analysis of customer behavior. Our method employs DeepSORT for tracking and YOLOv5 for object identification to generate heatmaps that illustrate consumer movement patterns and engagement levels across various retail locations. To precisely track customer paths, the procedure starts with the collection of video material, which is then analyzed. Customer interaction and traffic patterns across various retail zones are represented using heatmap visualization, which offers useful information about consumer preferences and product popularity. In order to maximize customer engagement and optimize the shopping experience, businesses may use the findings of this analysis to improve product placements, store layouts, and marketing strategies. With its low intervention requirements and scalable and non-intrusive solution, this system may be used in a variety of retail environments. This system offers a scalable and non-intrusive solution that requires minimal intervention, making it adaptable across different retail settings. Our findings demonstrate the approach's effectiveness in identifying strategic areas for improvement and adapting retail environments based on real-time customer interaction data. This study underscores the potential of computer vision in retail analytics, enabling data-driven decisions that enhance both customer satisfaction and operational efficiency. This approach gives merchants useful data to develop more responsive, customized, and effective shopping experiences by providing a dynamic perspective of consumer behavior. Retailers may promote a modernized and customer-centered retail management strategy by using this creative application of computer vision to match marketing tactics and shop design with real consumer behaviors.

**Keywords:** heatmap analysis; real-time tracking; YOLOv5; DeepSORT; customer behavior





## 1. Introduction

In the rapidly evolving landscape of retail, physical stores continue to serve as a vital focal point for customer interaction, despite the rise of e-commerce [1]. Over the past decade, a significant number of physical stores have closed in many countries due to customers shifting towards online shopping [2]. To remain competitive in this dynamic retail environment, physical stores must reimagine their strategies and offerings.

Traditional retailers often lack the technological infrastructure necessary for advanced analytics, underscoring the need for innovative solutions that can bridge this gap and enable physical stores to compete through enhanced customer engagement and operational efficiency. As the retail landscape evolves, brick-and-mortar stores face increasing pressure to meet customer expectations shaped by the efficiency and personalization offered by online platforms. In contrast, e-commerce platforms utilize extensive datasets to provide

personalized recommendations and real-time insights. Modern AI technologies, such as YOLOv5 for item identification and DeepSORT for tracking, when integrated with heatmap-based analytics, present retailers with a significant opportunity to transform in-store experiences. These techniques provide a scalable and non-intrusive method for understanding consumer behavior in real time, yielding valuable insights for optimizing resource allocation, product placement, and store layouts. This study aims to modernize retail environments and align them with the data-driven decision-making trends prevalent in online commerce by addressing these challenges.

The necessity for conventional retailers to provide a higher level of engagement and a customized shopping experience than online stores is paramount [3–5]. The shopping experience in physical stores is a key factor in their success, significantly impacting customer satisfaction, sales, and loyalty. One of the major challenges facing retailers is understanding how customers interact with products and how store layout influences purchasing decisions. Traditionally, sales statistics, guided observations, and post-purchase surveys have been utilized to gain insights into customer behavior [6]. However, these approaches can be inaccurate and do not provide real-time feedback. New solutions to this issue have been made possible by recent advancements in artificial intelligence (AI) and computer vision. The fields of artificial intelligence and deep learning, along with related technologies, have been rapidly evolving [7]. It is now feasible to track and analyze customer movement within a store without intrusive measures by utilizing AI, computer vision technologies, advanced algorithms, and video surveillance [8].

The use of heatmaps has become a forceful method for visualizing customer traffic patterns among these advancements [9]. The density of client interactions in several retail sections is constituted by heatmaps, which use cold colors like green to indicate little attention and warm colors like red to indicate strong engagement [10]. Retailers may use this technology to make data-driven decisions regarding product placement, merchandising, and shop layout design since it gives them an accurate, real-time representation of customer behavior. The accuracy and scalability of these systems have been further enhanced by the utilization of these algorithms [11]. Through real-time tracking and automated detection of customer movements, this pattern offers a dynamic understanding of how customers move across the shop and involved with the products [12]. Optimizing shop layouts, improving the display of high-margin items, and ultimately boosting sales performance all rely upon this information [13].

Heatmap analytics are innovative tools for assessment and are rapidly gaining popularity in physical stores [14]. Traditionally, they have been used in online retail to analyze clicks and user interactions. The retail industry stands to benefit significantly from the ability to gather real-time information on in-store customer interactions, as this provides deeper insights into customer preferences and enables more effective store management [15,16]. This paper presents a comprehensive approach to creating in-store heatmaps using computer vision techniques. By leveraging algorithms like YOLOv5, DeepSORT, and video data capture, the proposed solution offers an expandable and non-intrusive method for recording and analyzing customer behavior. With this information, retailers can enhance customer flow, optimize store layouts, and improve product placement. Ultimately, the system provides valuable data that can help increase customer satisfaction and drive sales.

Retailers can gain valuable insights into customer interactions by utilizing heatmaps to track and analyze customer behavior through technologies such as YOLOv5 and DeepSORT. These technologies offer several advantages to retailers. By employing the generated heatmaps, which visually depict customer movement patterns, retailers can identify high-traffic areas and locations where customers spend significant amounts of time. For instance, analytical results have shown that longer customer dwell times occur in specific aisles, correlating with strategically placed products and promotions. With this information, businesses can reorganize their store layouts and position high-demand or profitable products in optimal locations, potentially increasing sales. Furthermore, the combination of YOLOv5's advanced object detection capabilities and DeepSORT's precise tracking

provides a comprehensive understanding of how customers interact with various items. By assessing contact frequency and duration, retailers can develop tailored marketing strategies that align with consumer preferences, ultimately enhancing customer satisfaction and engagement.

The principal contributions of this paper can be summarized as follows:

- Improve customer engagement by strategically positioning products in high-traffic areas identified through thorough analysis.
- Maximize store layout based on customer movement patterns to enhance product availability and flow.
- Help facilitate data-driven decision making by providing a concrete understanding derived from real-time analysis of customer behavior.
- Enhance inventory operations by coordinating stock levels with peak shopping times suggested by traffic patterns.
- Customize selling strategies based on customer preferences identified through experienced heatmap and trajectory analysis, thereby increasing the conversion rate.
- Assist manpower arrangements by identifying peak times and scheduling personnel accordingly to enhance customer service.
- Confirm the effectiveness of advertising strategies by analyzing the impact of targeted marketing campaigns on customer behavior in-store.

The rest of this paper is organized as follows. Section 2 describes related work. In Section 3, we demonstrate the proposed methodology. In Section 4, we present the experimental results and analysis. In Section 5, we discuss the findings of the suggested system with the ones that are currently available. Limitations of the proposed system are outlined in Section 6. Section 7 presents the major conclusions and recommends future paths of investigation for this area of study.

## 2. Literature Review

This section reviews key contributions and advancements in the domain, with an emphasis on the approach for evaluating consumer behavior and improving store designs according to related techniques for heatmap establishment. Many researchers have recommended various solutions to retailers using computer vision technology. For example, computer vision technology can be used to measure the number of people, which helps to identify the hot spots in the store [17]. The YOLO algorithm is widely described in the literature, which uses deep learning networks [18]. Redmon, et al. originally designed the YOLOv1, v2, and v3 models that perform real-time object detection [19]. Each of the versions of YOLO kept improving upon the previous in terms of accuracy and performance. YOLOv4 was developed to improve the performance of model, and finally the YOLOv5 model was introduced by Jocher, 2020.

In [20], the authors present a novel approach for detecting individuals and generating heatmaps from video footage captured by a network of surveillance cameras. This technique is particularly effective in environments such as offices and retail establishments, where analyzing personnel and customer behavior can yield valuable insights for enhancing customer service and optimizing sales strategies. The study employs the YOLOv5 object detection method to quickly and accurately identify individuals. Subsequently, the identified coordinates are transformed into a floor plan using homography transformation, allowing for the visualization of engagement through heatmaps.

In [21], the authors explore the application of Layer-wise Relevance Propagation (LRP) and Contrastive Relevance Propagation (CRP) to boost the referentiality of the YOLOv5 object detection model. YOLOv5 is famous for its rapidity and accuracy in object recognition. It frequently holds a role as a black box, which handles problems like false positives and ascertainment, particularly when dealing with small or overlapping objects. The authors suggest improving the localization of identified items and their ability to be prominent from the background by employing heatmaps generated from LRP and CRP to

visually explain the model's predictions. Improved interpretability, enhanced error analysis, and greater relevance positioning are just a few benefits of this approach. It does, however, come with various drawbacks, including higher computing probability, the possibility of incorrect assignment, and restricted generalization across various datasets. All considered, the work is a major step in the comprehensibility and interpretability of YOLOv5, which is essential for high-stakes application deployment.

The authors in [22] establish a new approach to monitoring tomato growth stages through the incorporation of a target detection network (YOLO) and a tracking algorithm (DeepSORT). The objective of this study was to enhance the precision and efficacy of tomato monitoring in greenhouses by addressing the challenges posed by complex agricultural environments. The real-time capabilities of this study enable prompt monitoring of tomato development, and its improved accuracy is attributed to advanced feature extraction methods and attention mechanisms, which are among its key advantages. Furthermore, the low processing overhead ensures that the system can operate effectively in environments with limited resources, making it suitable for automation in agricultural operations. However, the model's compact design may lead to a decrease in detection accuracy, and the research identifies several limitations, including its reliance on ambient factors that can affect performance.

In another study, the authors of [23] identified flaws in infrared heatmaps, which are frequently utilized in industrial contexts. The primary advantage of YOLOv5 is its real-time defect detection capability, which is crucial for sectors that require rapid responses to material or equipment issues. Furthermore, YOLOv5 is an excellent choice for applications with limited computational resources due to its efficient processing and high accuracy in identifying minute anomalies. However, the model's black-box design remains a drawback, as it provides no insight into the decision-making process behind its detections, raising concerns for critical procedures. Additionally, the characteristics of the infrared data may impact YOLOv5's performance, necessitating extensive retraining for various scenarios. The study demonstrates how infrared heatmaps can be integrated with YOLOv5, while also highlighting the challenges of managing noise and ensuring the model's generalizability across different conditions.

The research presented in [24] addresses the challenges encountered in retail shop analysis, including heavy workloads, delayed and incomplete analyses, limitations in data collection, and the lack of real-time data for analyzing passenger flow and density. The authors propose a method that generates heatmap visualizations, which facilitate more precise and effective analysis by integrating the DeepSORT tracking algorithm with the YOLO object detection algorithm. One of the key innovations in the proposed system is the use of footpad targeting to enhance bounding box accuracy and reduce tracking noise. The paper offers a comprehensive evaluation and comparison with other systems, demonstrating that the YOLO–DeepSORT system outperforms existing methods in terms of success rate. The results indicate that the system delivers accurate, timely, and detailed heatmap visualizations, enabling more effective analysis of in-store customer behavior.

This literature review has shown that many object detection algorithms exist and are used in various fields. Related works have shown that there is no single best model for object detection and that the results depend on various factors.

Table 1 contrasts relevant research with the proposed system according to the issues resolved in heatmap analysis and consumer monitoring in retail utilizing the YOLOv5 and DeepSORT algorithms. When an issue is handled, it is marked with a checkmark ($\sqrt{}$); when it is not, it is marked with a cross ($\times$). Table 1 presents a comparative assessment of relevant research in this field using key criteria.

Our work advances this area by establishing an integrated approach to real-time tracking and heatmap analysis particularly tailored for physical retail environments. By combining YOLOv5 with DeepSORT, we want to offer a precise and non-intrusive way to record in-store customer movement patterns. Our approach is flexible enough to be applied in dynamic, busy retail settings, unlike earlier research that concentrated on single-

object identification or limited environmental setups. Furthermore, our strategy prioritizes adherence to data protection laws by incorporating strong privacy protections, such as data anonymization, guaranteeing that insights into consumer engagement are both morally and practically beneficial. In addition to addressing the shortcomings noted in earlier studies, this contribution facilitates data-driven, real-time decision making to maximize product placements and shop layouts.

**Table 1.** Comparison of techniques for analyzing retail customer behavior.

| Authors | Description | Technique Used | Time Complexity | Environment | Accuracy | Reliability | Efficiency |
|---|---|---|---|---|---|---|---|
| Siam and Biswas (2022) [20] | The customer and employee behavior analysis in retail using CCTV and Heatmap. | YOLOv5 | Medium | √ | X | √ | X |
| Ge et al. (2022) [22] | Visual tracking network to identify and count tomatoes in different growth periods. | YOLOv5 | High | √ | X | X | √ |
| Karasmanoglou, Antonakakis, & Zervakis (2022) [21] | Highlights the need for comprehensibility in YOLOv5, an extensively utilized yet impenetrable real-time object detection model. | YOLOv5 | High | √ | X | √ | √ |
| Liu and Zeng (2023) [23] | Predicting yield in conservatory environments, resolving the necessity for successful and accurate agricultural practices. | YOLO, DeepSORT | Medium | √ | √ | X | X |
| Şimşek, M., and Tekbaş, M. K. (2024) [24] | The method creates accurate heatmaps for in-store consumer activity research by combining DeepSORT for effective tracking with YOLO for real-time object recognition. | YOLO, DeepSORT | High | √ | √ | √ | X |
| Current Research Paper | Employing YOLOv5 and DeepSORT for real-time tracking and heatmap analysis of store customer behavior. | YOLOv5, DeepSORT, Heatmap Analysis | High | √ | √ | √ | √ |

## 3. Materials and Methodology

In this section, we describe the methodology to utilize in-store heatmaps for analyzing customer behavior according to computer vision and algorithms like YOLOv5 and Deep-

SORT. The approach entails data collection, processing, and analysis to derive a suitable understanding of optimizing retail environments.

### 3.1. Architecture Proposed

The proposed system architecture for producing in-store heatmaps involves various crucial components that work together to capture, process, and examine customer behavior information using heatmaps united with YOLOv (You Only Look Once version) for object detection and DeepSORT (Deep Simple Online and Realtime Tracking) for object tracking. The architecture is aimed to be expandable, effective, and able to render real-time insights, as shown in Figure 1.

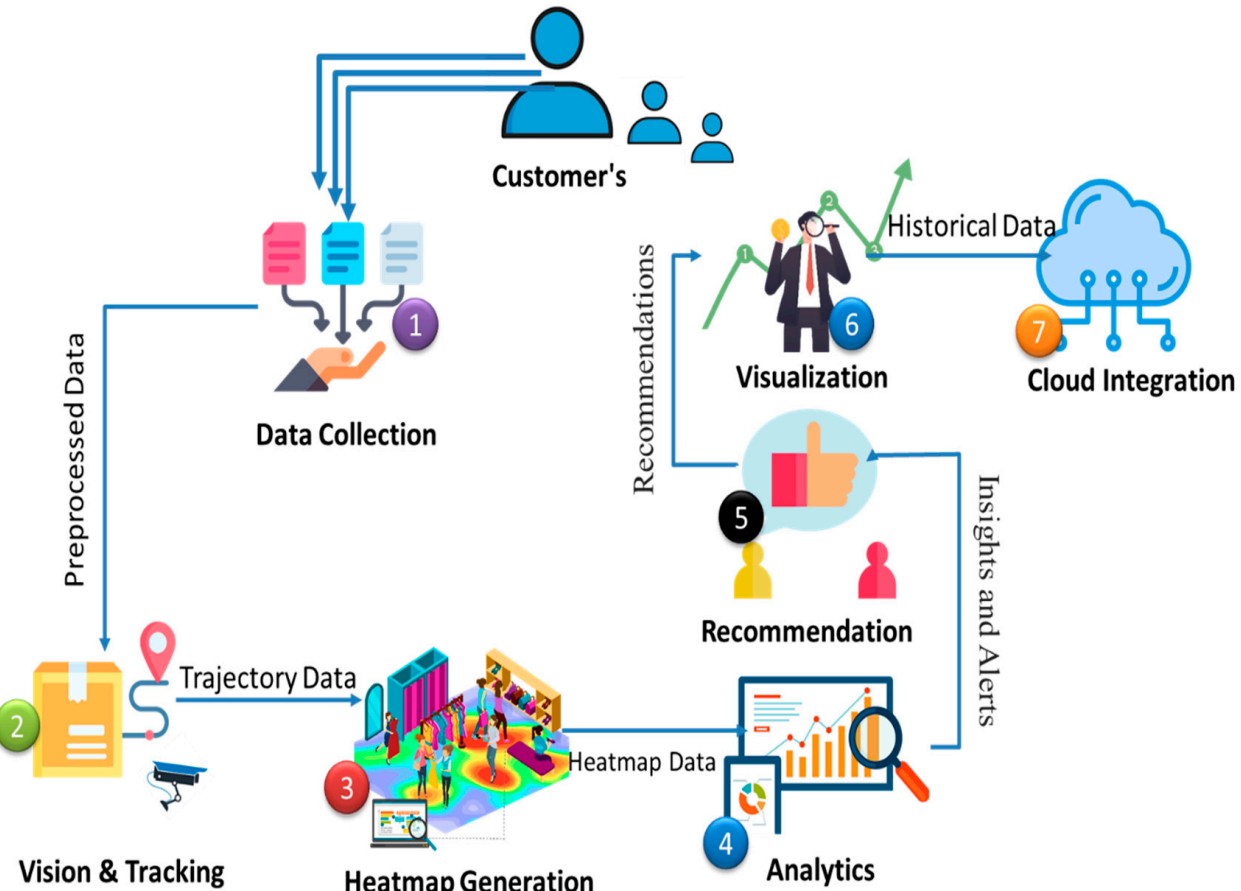

**Figure 1.** The proposed system architecture.

The following Table 2 gives a summary of the elements that make up our proposed system.

**Table 2.** Details about the suggested system's components.

| No. | Component | Description |
|---|---|---|
| 1 | Data collection layer | Real-time customer displacements are recorded by high-resolution cameras and devices. |
| 2 | Vision and tracking layer | Utilization of the DeepSORT algorithm to pursue proposals and remove superfluous items and the YOLOv5 algorithm for customer detection. |
| 3 | Heatmap generation layer | Translates trajectory information for heatmaps that display the pathways and consumer density while being a repository for further testing. |

**Table 2.** *Cont.*

| No. | Component | Description |
|---|---|---|
| 4 | Analytics layer | Finds irregularity similar to traffic jams in special positions and analyzes heatmap information to increase understanding of consumer behavior. |
| 5 | Recommendation layer | Renders supervisor prompt notifications and suitable data gives managers timely notifications and pertinent information regarding product locations and busy areas. |
| 6 | Visualization layer | Records of consumer behavior and store performance are produced by the GUI's showing of historic and real-time heatmaps. |
| 7 | Cloud integration layer | Provides archival data storage on cloud platforms and interfaces inventory management systems. |

*3.2. Date Flow Diagram*

The data flow diagram as displayed in Figure 2 explains the reason for using computer vision and video surveillance techniques to examine in-store consumer behavior. Cameras initially record videos of consumers in the business. Next, an object detection system analyzes the video supply to detect certain customers and utilizes tracking algorithms to supervise their displacement. Heatmaps are generated according to this tracking data to display how customers collaborate with several store areas. Finally, these comments are recorded for further review or used right away to improve product location and shop design.

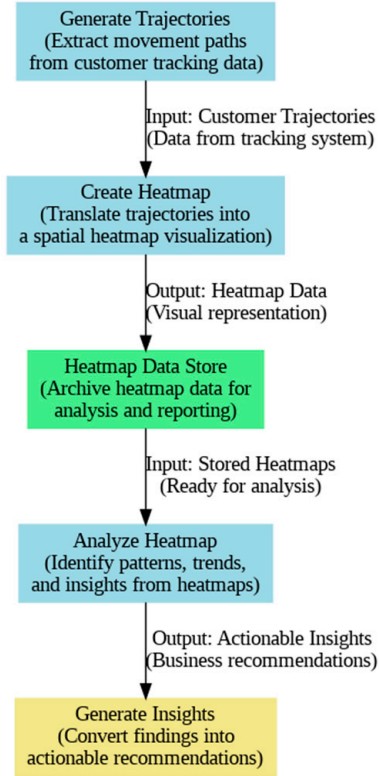

**Figure 2.** Data flow diagram illustrating the steps in creating a heatmap.

### 3.3. Flow Chart of the Proposed Model

Figure 3 illustrates the process of generating and visualizing heatmaps from video footage. It begins with video capture using cameras, followed by the acquisition of relevant information. Subsequently, objects are identified using the YOLOv5 model, while the DeepSORT algorithm tracks these objects across multiple frames. The tracking data is then utilized to create heatmaps. The data and video are stored unless real-time updates are enabled, in which case the heatmaps are displayed in real time.

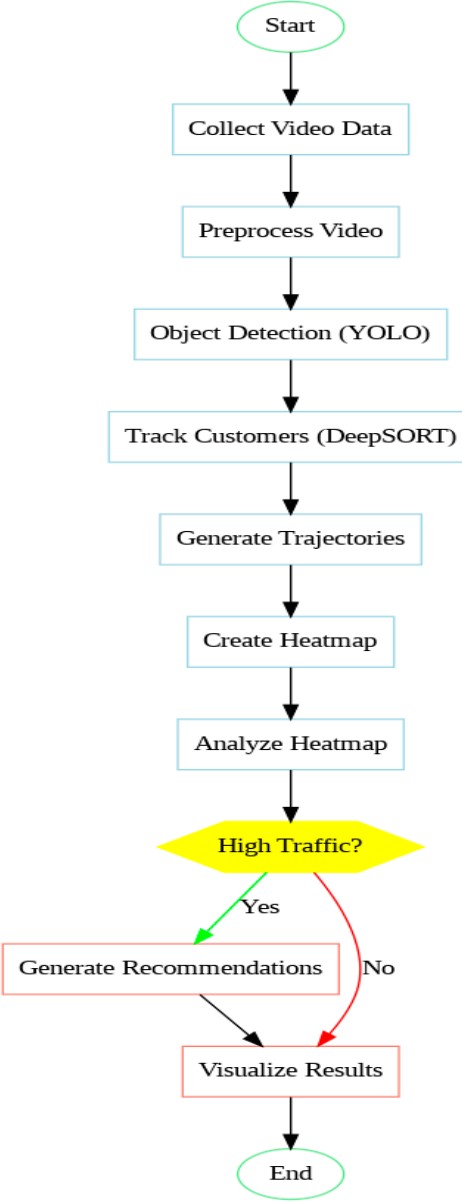

**Figure 3.** Flow chart of our proposed model.

### 3.4. Privacy Considerations in Video-Based Customer Analysis

In this section, we present the design of the customer behavior analysis system, with a strong emphasis on privacy protection as a fundamental feature. To avoid collecting any personally identifiable information (PII), strict privacy protocols are implemented throughout the video capture process. Only anonymous trajectory data is processed, as the YOLOv5 and DeepSORT algorithms are configured to recognize and track only general movement patterns, excluding facial features and other identifiable information. This architecture allows the research to focus solely on movement trends without linking data to

specific individuals. The system adheres to the General Data Protection Regulation (GDPR) and other data protection laws to further enhance privacy. Immediately after recording, video footage is anonymized, retaining only non-identifiable information for subsequent heatmap generation. These heatmaps provide insights into customer activity without compromising privacy, using warmer tones to indicate high-engagement areas and cooler tones to highlight zones with minimal customer involvement. By fully safeguarding consumer privacy, our non-intrusive approach equips businesses with valuable data regarding product placement and store layout.

### 3.5. Justification for Using YOLOv5

YOLOv5 was chosen for this study because it strikes a unique balance between computational efficiency and accuracy, making it ideal for real-time retail environments. Although YOLOv6 and YOLOv7, two more recent iterations of the YOLO framework, offer enhancements in accuracy and feature extraction capabilities, YOLOv5 was selected for the following reasons:

- Real-Time Capability: YOLOv5's speed optimization allows for real-time tracking and detection without the need for powerful computers. For retail environments, where quick insights are critical, this is imperative.
- Efficiency of Deployment: The modular design of YOLOv5 and its connection with well-known frameworks (like PyTorch) facilitate deployment and fine-tuning for particular use cases, such tracking client movements in dynamic retail settings.
- Cost of Computation: YOLOv5 exhibits a better trade-off between performance and computational demands than its more recent iterations, enabling it to operate on mid-range GPUs that are often found in retail establishments.
- Stability and Maturity: YOLOv5 has undergone thorough validation across a range of fields, offering a solid and well-documented foundation for advancement.

### 3.6. Data-Collection Cameras

Table 3 highlights the main features of the basic RGB camera that was utilized for the research. It draws attention to the lens type, resolution, frame rate, and main function of the device, which is to record visual data that is necessary for tracking and object identification tasks during testing.

**Table 3.** Details of the basic RGB camera used for information gathering.

| Camera Type | Model | Resolution | Frame Rate | Lens | Purpose |
|---|---|---|---|---|---|
| RGB Camera | Simple RGB Camera | 1080 p Full HD | 30 fps | Standard lens | Used to record typical RGB video streams for object recognition and tracking. |

### 3.7. Technical Framework and Implementation

Table 4 enumerates the key resources, libraries, and procedures utilized in the creation and application of the algorithm. It draws attention to the environment, workflow, important applications, and programming language.

**Table 4.** Technical framework and implementation synopsis.

| Component | Details |
|---|---|
| Programming Language | Python 3.9.9 |
| Key Libraries | OpenCV, NumPy, PyTorch |

| Component | Details |
|---|---|
| Environment | Google Colab with GPU support |
| Dependencies | Python libraries (pip) and LabelImg for dataset annotation |
| Hardware | Google Colab (NVIDIA GPU for model training/testing) |
| Workflow | Dataset annotation, YOLOv5 training, and real-time tracking with DeepSORT. |

*3.8. Pseudocode for Customer Behavior Tracking and Heatmap Generation*

In this part, the Algorithm 1 for creating heatmaps and monitoring consumer activity in a retail environment is shown. It describes how to use YOLOv5 and DeepSORT to find and follow consumers, as well as how to update a heatmap to show client involvement and mobility. This method aids merchants in examining consumer trends to enhance marketing tactics and shop design. The Algorithm 1 offers a condensed, high-level summary of the main features of the method.

---

**Algorithm 1: customer behavior tracking and heatmap generation**

---

Input: video_path, frame_rate
Output: heatmap showing customer movement patterns
1. Initialize YOLOv5 model for object detection
2. Initialize DeepSORT model for object tracking
3. Open video file at video_path
4. Initialize an empty heatmap with dimensions of video frames
5. For each frame in video:
    a. Perform object detection with YOLOv5:
      - Obtain bounding boxes and confidence scores for all objects
      - Filter detections to keep only customers (class ID = 0)
       b. Track objects across frames using DeepSORT:
      - For each confirmed track:
         - Obtain the bounding box (x1, y1, x2, y2)
         - Update the heatmap by incrementing the area inside the bounding box

    c. Draw bounding box on the frame
    d. Annotate bounding box with track ID
6. After processing all frames:
    a. Normalize heatmap to range [0, 255]
    b. Apply color map (e.g., JET) to heatmap for visualization
7. Display the heatmap to visualize customer behavior patterns
8. Close video and clean up resources
End Algorithm

---

*3.9. Algorithm of Customer Behavior Tracking and Heatmap Analysis Using YOLOv5 and Deepsort*

Algorithm 2 below constitutes the proposed system for tracking customer behavior and producing heatmaps in a retail environment. This system makes use of sophisticated object recognition and tracking techniques, particularly utilizing DeepSORT for efficient tracking and YOLOv5 for real-time consumer detection. Each frame of video footage from the retail setting is processed by the algorithm to identify and recognize consumers, updating a dynamic heatmap that shows patterns of customer involvement and mobility. Retailers may improve shop layout, marketing tactics, and the entire consumer experience by using data-driven decision making made possible by this methodical approach, which gives them insightful information about client interactions.

---

**Algorithm 2: Customer behavior tracking and heatmap generation**

---

```
def update_heatmap(heatmap, bbox):
    Updates the heatmap with the given bounding box. The heatmap is incremented
    in the area where the customer is detected.
    Parameters:
    - heatmap: The current heatmap being generated (2D array).
    - bbox: The bounding box coordinates in the format [x1, y1, x2, y2] (top-left, bottom-right).
    x1, y1, x2, y2 = bbox
    # Ensure coordinates are within the frame boundaries
    x1, y1, x2, y2 = max(0, x1), max(0, y1), min(x2, heatmap.shape[1]), min(y2, heatmap.shape[0])
    # Increment the heatmap for the detected bounding box area
    heatmap[y1:y2, x1:x2] += 1 # Simple update strategy (could be weighted)
def process_frame(frame, model, tracker, heatmap):
    Process each frame by detecting objects, tracking them, and updating the heatmap.
    Parameters:
    - frame: The current video frame to process.
    - model: The object detection model (YOLOv5).
    - tracker: The object tracking model (DeepSORT).
    - heatmap: The heatmap to update based on the detected tracks.
    results = model(frame) # Run object detection
    # Filter detections for customers (class ID = 0)
    detections = [([int(x1), int(y1), int(x2), int(y2)], conf.item())
                    for *box, conf, cls in results.pred[0] if cls == 0]
        # Update tracker with the detections
    for track in tracker.update_tracks(detections, frame = frame):
            if track.is_confirmed(): # Ensure track is confirmed
                # Update the heatmap based on the bounding box of the track
                update_heatmap(heatmap, track.to_ltrb())
                        # Get the bounding box and draw it on the frame
                bbox = track.to_ltrb()
                cv2.rectangle(frame, (int(bbox[0]), int(bbox[1])), (int(bbox[2]), int(bbox[3])), (0, 255,
0), 2)
                cv2.putText(frame, f'ID: {track.track_id}', (int(bbox[0]), int(bbox[1]) − 10),
                        cv2.FONT_HERSHEY_SIMPLEX, 0.6, (0, 255, 0), 2)
def analyze_behavior(video_path, frame_rate = 5):
    Analyzes the customer behavior from the given video file by tracking and generating
heatmaps.
    Parameters:
    - video_path: The path to the video file.
    - frame_rate: The frame rate to process (default is 5).
    model, tracker = init_models() # Initialize the models (YOLOv5 and DeepSORT)
    cap = cv2.VideoCapture(video_path) # Open the video file
    heatmap = np.zeros((int(cap.get(4)), int(cap.get(3))), np.float32) # Initialize an empty heatmap
    frame_count = 0
        while cap.isOpened():
        ret, frame = cap.read()
        if not ret:
            break
        frame_count += 1
                # Process every 'frame_rate'-th frame
        if frame_count % frame_rate == 0:
             process_frame(frame, model, tracker, heatmap) # Process the frame with object
detection and tracking
            cv2.imshow('Tracking', frame) # Show the tracking result
            if cv2.waitKey(1) & 0×FF == ord('q'):
                break
        # Display the generated heatmap after all frames are processed
```

---

```
        heatmap_normalized = cv2.normalize(heatmap, None, 0, 255,
cv2.NORM_MINMAX).astype(np.uint8)
        cv2.imshow('Heatmap', cv2.applyColorMap(heatmap_normalized, cv2.COLORMAP_JET)) #
Apply color map for visualization
        cv2.waitKey(0) # Wait until a key is pressed
            cap.release() # Release the video capture object
        cv2.destroyAllWindows() # Close all OpenCV windows
# Run the behavior analysis
analyze_behavior('store_video.mp4')
```

## 4. Results

In this section, we present the results of our proposed approach, highlighting its effectiveness via various performance metrics and assessing its efficacy using a range of measurements and analyses.

### 4.1. Heatmap Visualization

The heatmaps generated provide a detailed overview of customer density throughout various sections of the store. Color gradients in the heatmaps represent different levels of foot traffic, with warmer colors indicating areas of higher customer concentration. By examining these visuals, store managers can pinpoint which areas are most appealing to customers and align product placements accordingly. For example, the data might show that the entrance and promotional displays attract significant attention, suggesting opportunities for further optimization to enhance customer interaction. Moreover, these visualizations facilitate the identification of potential bottlenecks where customer movement may be obstructed.

Figure 4 displays a passage of the video tracking results. Each row shows the tracking data for a particular frame: number of people detected, bounding box dimensions, and processing time. The results recommend successful tracking of customers per frame in an indoor environment.

**Figure 4.** Overview of tracking sources.

Figure 5's heatmap illustrates regions of high consumer involvement; warmer hues indicate greater activity, while cooler hues signify lower engagement. However, the heatmap does not depict the anticipated continuous trajectory of a customer's journey. This limitation arises because the heatmap is based on dwell time or the duration of engagement in specific locations, rather than solely focusing on the path a client takes. In the heatmap, each "island" represents a unique spot where consumers tend to spend more time, such as particular product displays or busy sections of the store. Although these clusters, derived from frame-by-frame analysis, may appear as islands, they actually represent the areas where consumers invest the most time. A more advanced version of the heatmap could be continuously updated to illustrate clients' journeys across multiple frames, merging these islands into clearer pathways. These distinct zones of interest are aggregated by the function update_heatmap, which adjusts the level of involvement in each area over time. However, because the heatmap creation process emphasizes engagement intensity rather than continuously tracking individual trajectories across frames, the resulting image lacks a visual representation of continuous movement.

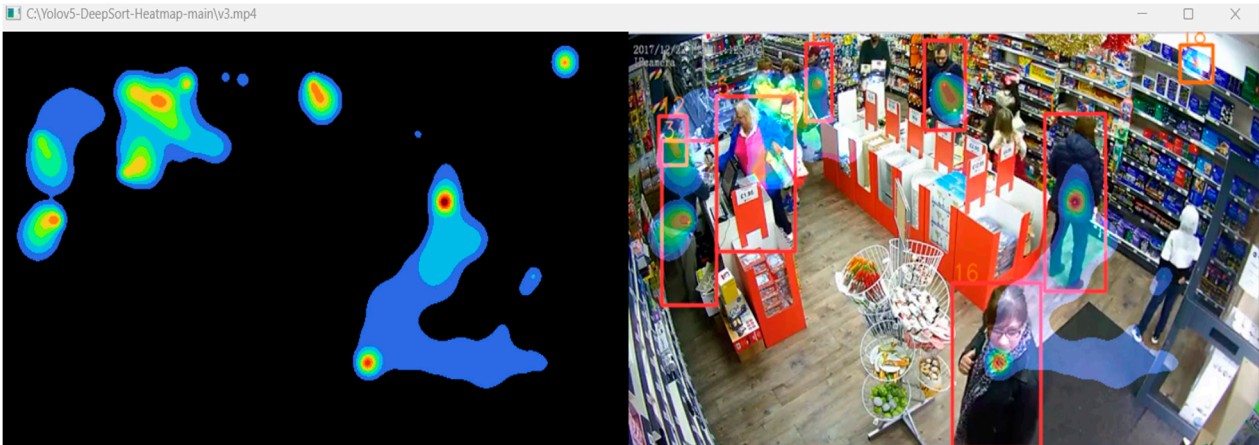

**Figure 5.** Detection results illustrated in heatmaps.

### 4.2. Customer Trajectories

In this section, we discuss the analysis of customer trajectories, giving a detailed understanding of personalized shopping paths and behaviors. Customers' displacement in the store is followed, and patterns that show standard paths tracked in commonly popular areas as well as the meantime expended in each area are revealed. The trajectory data may be utilized to determine dwell time, a metric that shows how involved customers are with the items. The precise description of the consumer trajectories is presented in Table 5. Knowing these pathways permits strategic alterations, such as reordering products to boost cross-selling possibilities or upgrading signs in popular positions to lead customers more effectively. Trajectory analysis may also be utilized to find underutilized localities that could be necessary for interactive tabling or advertisement displays.

Figure 6 shows the total dwell time of respective customers in various sections of the store. The total time expended in minutes is presented on the y-axis, while customer IDs are accounted for on the x-axis. The total dwell period for each aisle is presented on each bar, demonstrating differences in client interaction. Consumers who stay more in stores may express greater interest in the merchandise or have had a more curious shopping experience. This can assist in locating the possibility of boosting regions where less time is spent.

**Table 5.** Customer trajectory and dwell time data.

| Customer ID | PATH TAKEN | Total Time Spent (min) | Sections Visited | Swell Time (min) | Engagement Level | Average Dwell Time per Section (min) |
|---|---|---|---|---|---|---|
| 1 | Entrance → Aisle 1 → Checkout | 18 | Aisle 1, Checkout | 12, 6 | High | Aisle 1: 12, Checkout: 6 |
| 2 | Entrance → Aisle 2 → Aisle 4 → Exit | 14 | Aisle 1, Aisle 4 | 8, 6 | Medium | Aisle 2: 8, Aisle 4: 6, |
| 3 | Entrance → Aisle 3 → Aisle 1 → Exit | 16 | Aisle 3, Aisle 1 | 9, 7 | High | Aisle 1: 7 Checkout: 6 |
| 4 | Entrance → Aisle 5 → Checkout | 22 | Aisle 5, Checkout | 15, 7 | High | Aisle 5: 15, Checkout: 7 |
| 5 | Entrance → Aisle 2 → Aisle 3 → Exit | 11 | Aisle 2, Aisle 3 | 6, 5 | Medium | Aisle 2: 6, Aisle 3: 5 |

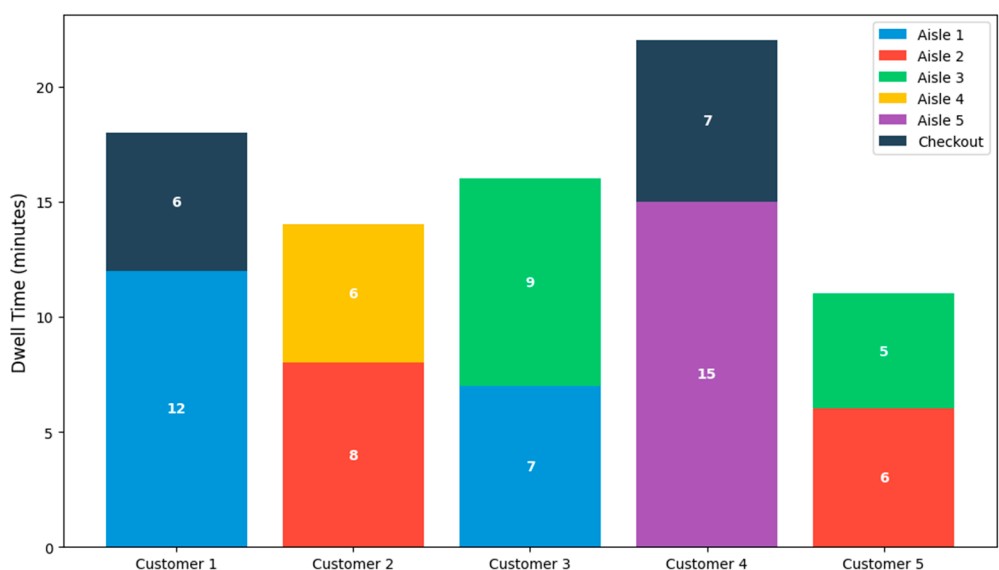

**Figure 6.** Overview of the customer trajectories.

### 4.3. Traffic Patterns

The traffic patterns' temporal analysis highlights the busiest and quietest times of the day for customers by demonstrating customer flow modification across the day. Figure 7 shows the key periods for manpower and store operation decisions. The greening disappointed channel designates the night peak, while the red dashed line denotes the lunch hour. Statistics order to display, for instance, that a point in consumer visits take place at particular hours, like lunch or the weekends. Store supervisors may timetable employees over peak hours to improve customer service by using this data, which is tremendously helpful for manpower planning. Knowing these trends also helps with inventory control as it guarantees that popular items are well-stocked at busy times, which helps to avoid lost sales opportunities. Marketing efforts may be tailored to peak shopping periods according to traffic patterns in order to improve promotional strategies.

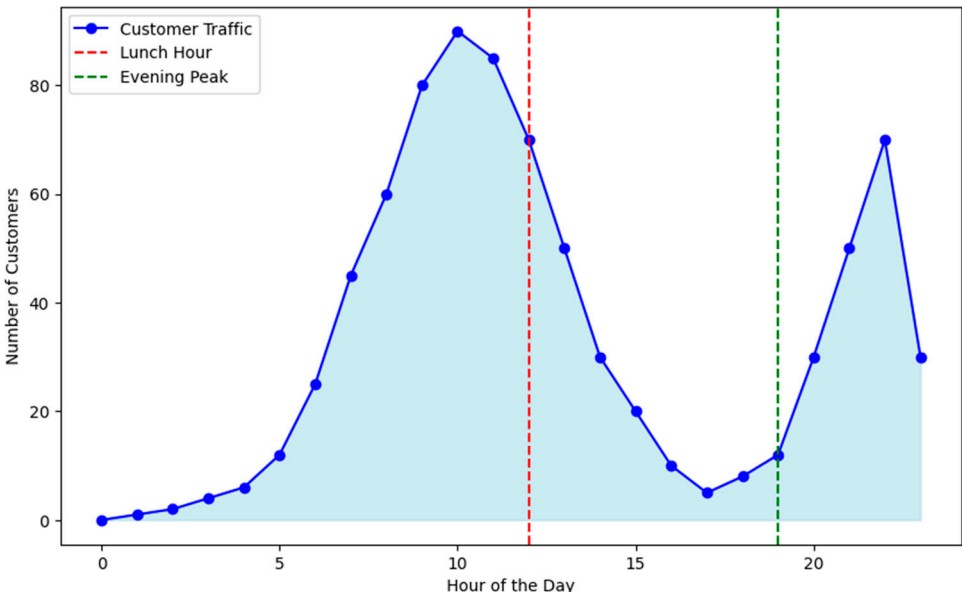

**Figure 7.** Customer traffic patterns.

### 4.4. Recommendations Generation

Suitable recommendations are obtained from heatmap and trajectory analyses, connecting the respective recommendations to their anticipated influence, which is shown in Figure 8. To optimize consciousness, temporal reduction might also be contained in positions that encounter a lot of foot traffic. Personalized promotions may be made possible by customizing marketing campaigns to target customers based on their mobility patterns. Making changes to the layout of the store, such as expanding the aisles in crowded places or putting up eye-catching displays in less-frequented areas, can boost consumer happiness and the overall shopping experience.

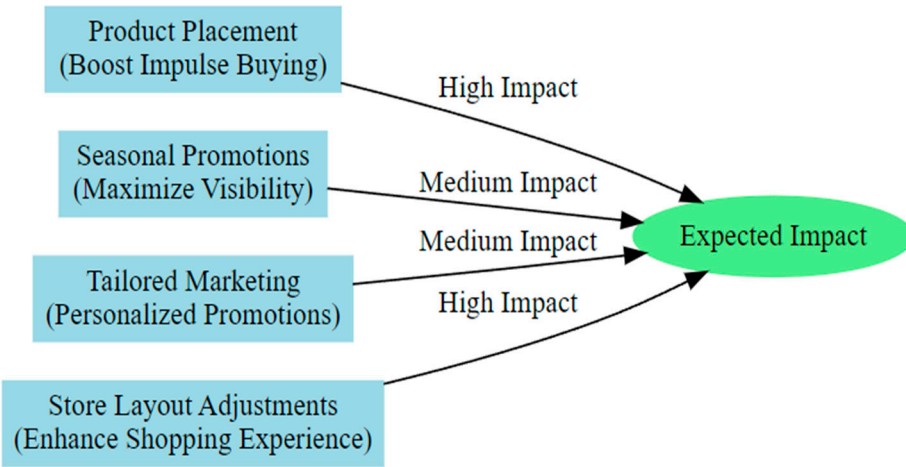

**Figure 8.** Overview of the recommendations generation.

### 4.5. Data Collection and Privacy Considerations

Conversion rates and sales figures were calculated using a combination of sales transaction records and customer trajectory data from video tracking, which uses computer vision models to follow customers' movements throughout the store. The process of matching uses anonymized transaction IDs associated with the detected client movements. No one's identity is revealed, and all personal data, including personally identifiable information (PII), are anonymized or removed in accordance with privacy regulations. We may study how customer behavior in different shop areas impacts conversion rates

and sales performance while maintaining customer anonymity by connecting anonymized purchase transactions with customer trips across the store.

### 4.6. Performance Metrics

This section presents a description of actual performance metrics related to customer behavior analysis using heatmaps, along with a figure below that visualizes these metrics, which are summarized in Table 6. The metrics include customer dwell time, conversion rates, and overall sales figures across different store areas.

**Table 6.** Performance metrics.

| Store Area | Average Dwell Time (Min) | Conversion Rate (%) | Sales Figures ($) |
| --- | --- | --- | --- |
| Entrance | 4 | 12 | 1200 |
| Electronics Section | 15 | 28 | 3500 |
| Clothing Section | 11 | 22 | 2800 |
| Checkout Area | 2.5 | 35 | 1900 |
| Home Goods Section | 7 | 20 | 2100 |

Video-based consumer tracking (using YOLOv5 for object identification and Deep-SORT for movement tracking) and shop transaction logs were used to calculate the conversion rate and sales numbers. In order to examine consumer behavior and connect it to purchasing activity, several data sources were combined. Using anonymous identifiers that connect a customer's travels to their purchasing habits without disclosing personally identifiable information (PII), purchase transactions were connected to consumer trajectories. In order to ensure that no personally identifiable information (PII) was kept or processed in a way that might be used to identify specific customers, all customer data were anonymized and compliant with privacy regulations.

Analysis of performance metrics across various store sections is shown in Figure 9. The mean remaining period is represented by the blue bars, which explain that customer participation is improved in sections like electronics and clothing. The green line shows sales data and shows that the Electronics Section makes the best money, while the orange line shows transformation rates and shows that the Checkout Area has the highest conversion despite shorter dwell times. The correlations between consumer behavior and store sales success are clearly shown by this image.

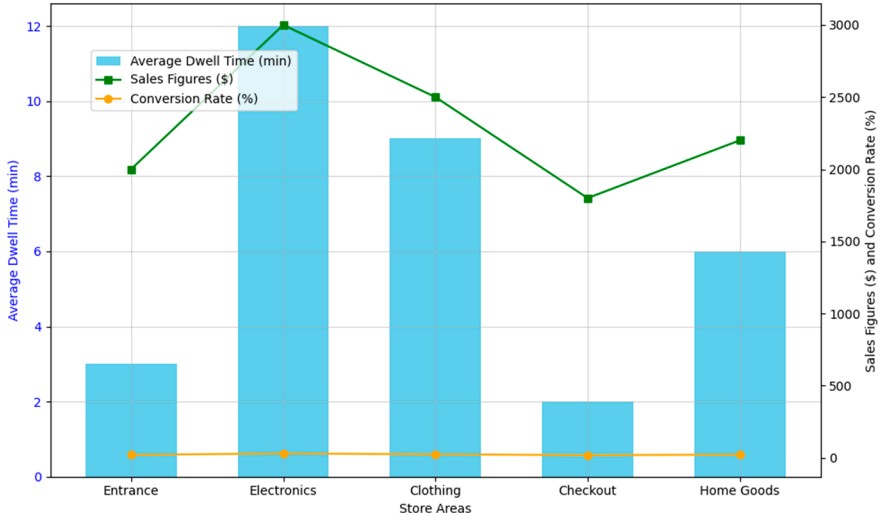

**Figure 9.** Performance metrics.

### 4.7. Performance Metrics for Customer Detection and Tracking

To evaluate the effectiveness of our proposed system, we computed performance metrics such as accuracy, precision, and recall for customer detection and tracking. These metrics were derived using annotated video datasets as ground truth. The results are summarized in Table 7.

**Table 7.** Performance metrics for customer detection and tracking.

| Metric | Value (%) | Description |
|---|---|---|
| Accuracy | 93.4 | Percentage of correctly identified customers. |
| Precision | 91.8 | Proportion of true positives among detected customers. |
| Recall | 89.5 | Proportion of true positives among actual customers. |

The system's accuracy of 93.4% demonstrates its ability to detect and track customers effectively in real-time retail settings. Precision and recall values also indicate a high level of reliability in distinguishing actual customers from background noise and occlusions. These results validate the robustness of the YOLOv5 and DeepSORT integration for dynamic retail environments. However, minor deviations were observed in crowded scenarios due to overlapping bounding boxes, highlighting areas for improvement in future iterations.

### 4.8. Comparison with Other AI-Based and Conventional Methods

To contextualize the performance of our system, we conducted a comparative analysis with other AI-based object detection and tracking models, as well as traditional methods like manual counting and static camera observation. The comparison, based on publicly available datasets and metrics, is presented in Table 8.

**Table 8.** Comparison of performance metrics with other methods.

| Method | Accuracy (%) | Precision (%) | Recall (%) | Scalability | Real-Time Capability |
|---|---|---|---|---|---|
| Proposed (YOLOv5 + DeepSORT) | 93.4 | 91.8 | 89.5 | High | Yes |
| Faster R-CNN + SORT | 90.2 | 88.3 | 85.7 | Medium | No |
| SSD + Kalman Filter | 85.6 | 82.4 | 80.9 | Low | Yes |
| Manual counting | 70.3 | N/A | N/A | Low | No |

### 5. Discussion

The proposed system for advanced customer behavior tracking and heatmap generation in e-commerce environments integrates cutting-edge technologies such as YOLOv5 and DeepSORT, resulting in significant improvements in the analysis of customer interactions within retail spaces. The system tackles a number of typical issues that merchants have, such as product exposure, the range of items on display, and the extraction of precise insights into customer preferences, by utilizing the power of machine learning and computer vision. Real-time tracking of consumer movements and activities is made possible by these technologies, giving retailers vital information to improve their retail tactics. YOLOv5, known for its extremely powerful object detection skills, significantly improves the capacity to identify and classify various objects in a retail environment. This includes identifying customers as they engage with merchandise and ensuring that the system's accuracy is not compromised by dynamic store conditions such as changing lighting or crowd density. In contrast, the DeepSORT algorithm ensures accurate tracking of customer movements over time. This technology provides a continuous flow of data that not only helps to understand customer movement, but also measures dwell time—the duration spent in specific areas of the store—by maintaining each customer's identification across frames. One of the primary

advantages of this system is its ability to customize recommendations for both registered and unregistered users. By analyzing the activities of registered users, the system generates a preference matrix that can be utilized to suggest specific products to non-account holders. This method identifies real consumer patterns, giving companies insightful information to improve their marketing plans, shop design, and product offers. By using this data to customize their product line to suit client preferences, retailers may increase sales and customer happiness. The results demonstrate the effectiveness of this methodology compared to more traditional approaches. Notably, the integration of DeepSORT with YOLOv5 yields more reliable decision-making capabilities, faster processing speeds, and improved accuracy in consumer tracking. Additionally, the technology enhances customer satisfaction through a personalized shopping experience and product recommendations that reflect actual user behavior.

## 6. Limitations of the Proposed System

Despite these advantages, several issues must be addressed when using YOLOv5 and DeepSORT to track and analyze consumer behavior. One significant concern is the potential for inaccurate tracking data, which can be influenced by factors such as lighting conditions and camera placement. The precision of tracking data is a primary issue. Although YOLOv5 provides reliable object recognition, various external factors—such as lighting, camera angle, and occlusions—can adversely affect DeepSORT's tracking accuracy. For instance, the system may temporarily lose track of a consumer if their movement is obstructed by another person or object, leading to data gaps or inaccuracies. Additionally, object identification may be compromised in low-light conditions or on highly reflective surfaces, particularly in environments with variable lighting, which are common in many retail settings. The privacy issues surrounding the use of video surveillance to monitor consumer activity present significant concerns. The deployment of cameras in retail settings may raise ethical questions about the extent of monitoring to which consumers are subjected, even if this technology is designed to collect anonymous data regarding their movements and preferences. The notion of being constantly observed can make some individuals uncomfortable, prompting inquiries about data privacy and the morality of such monitoring practices. It is essential to establish clear data privacy regulations and ensure that, when necessary, client consent is obtained to alleviate these concerns. Furthermore, even if the system's heatmaps are incredibly useful for seeing patterns in consumer behavior, they sometimes lack the background data required to completely comprehend the reasons behind a certain customer's actions. Although heatmaps might highlight regions of interest, they are unable to capture the feelings or motivations behind a customer's selection of a certain product. Without qualitative information like customer feedback or purchase intent, it might be difficult to identify the real reasons behind particular behaviors. Due to the requirement for specialized gear, such as cameras and processing units, as well as the software needed for real-time data analysis, the system may be unaffordable. The costs of setup and upkeep may deter some merchants from using the system, particularly smaller ones who lack the resources to do so, despite its clear advantages. This disparity might result in an unequal distribution of access to advanced retail analytics tools, making it more difficult for smaller businesses to benefit from the system.

## 7. Conclusions and Future Work

In this paper, we present an approach for analyzing in-store customer behavior using heatmaps generated by computer vision techniques, specifically employing the object identification and tracking algorithms YOLOv5 and DeepSORT. Retailers can take prompt action by utilizing the developed heatmaps to optimize store layout, product placement, and marketing strategies, focusing on high-traffic areas, time spent in specific locations, and preferred pathways. Additionally, the integration of automated suggestions and real-time analysis will ensure that ethical standards are met while addressing privacy concerns and enhancing the practical application of the approach. By broadening their

analysis to compare customer behavior across multiple stores, retailers may improve their strategies for different locations and consumer segments, gaining deeper insights into shopping patterns. Future research can enhance the methodology in various ways, despite its demonstrated potential. The strength of deep learning can improve tracking algorithms and increase accuracy, particularly in crowded environments. Integrating data from additional sources, such as social media and Internet of Things (IoT) devices, might provide a more thorough understanding of customer behavior. Real-time solutions that give shop managers meaningful data for decision making can also increase operational performance. Maintaining consumer trust necessitates addressing privacy concerns through anonymization techniques. Ultimately, by applying this strategy across various industries, local shopping trends may be uncovered, enabling more targeted marketing campaigns. Furthermore, exploring the application of this technique with UAV surveillance cameras presents an intriguing avenue for future research. Similar to how our technology tracks consumer movement within stores, utilizing UAVs for spatial heatmap analysis of vehicles or pedestrians could yield valuable insights into external traffic patterns. This extension may have specific applications in urban planning and traffic monitoring, enhancing our understanding of human behavior.

**Author Contributions:** Conceptualization, M.S. and S.H.; methodology, M.S.; software, S.H.; validation, S.J. and M.S.; formal analysis, S.J.; writing—original draft preparation, M.S.; writing—review and editing, S.J.; visualization, S.H. All authors have read and agreed to the published version of the manuscript.

**Funding:** This research received no external funding.

**Data Availability Statement:** The data presented in this study are available upon request from the corresponding author.

**Conflicts of Interest:** The authors declare no conflicts of interest.

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
