# Peer review of "Advanced Customer Behavior Tracking and Heatmap Analysis with YOLOv5 and DeepSORT in Retail Environment"

_electronics, doi:10.3390/electronics13234730_

Round 1
Reviewer 1 Report
Comments and Suggestions for Authors
Basically my problem is that some parts of paper are too short, and the results are not enough detailed.
Comments:
The 3. chapter should be named as "Materials and Methodology".
I think, the algorithm should be moved to "Materials and Methodology", not "Results".
Which software was used for the algorithm?
On Figure 5 I see the heatmap, but I dont see, how the algorithm can distinguish the heat from human body and heat from electronical devices.
Please write, which thermal and normal camera was used for the test.
The chapters 5-6-7. are too short, please extend it.
The format of references seems not MDPI standard, please check it.
Comments on the Quality of English LanguagePlease revise it with native English speaker.
Author Response
Response to Reviewer 1 Comments –Round 2
Thank you for your extremely useful comments and suggestions. Based on your feedback, the paper has been thoroughly revised. We have made every effort to address all comments in the updated manuscript. The most significant changes are highlighted in yellow in the revised document. A point-by-point response to each comment is provided below:
Comment 1: - The 3. chapter should be named as "Materials and Methodology".
Response: Thank you for your valuable feedback. Thank you for pointing this out. The title of Chapter 3 has been updated to "Materials and Methodology" to align with standard conventions. This change is reflected on page 6.
Comment 2: - I think, the algorithm should be moved to "Materials and Methodology", not "Results".
Response: Thank you for your valuable feedback. Thank you for the suggestion. The description of the algorithm has been moved from the "Results" section to the newly named "Materials and Methodology" section. This provides a clearer and more logical structure to the paper. This change is reflected on pages 9-11.
Comment 3: Which software was used for the algorithm?
Response: Thank you for your valuable feedback. Section 3.7, titled Technical Framework and Implementation, has been added to the "Materials and Methodology" section on pages 9 and 10. This section provides detailed information about the software environment and dependencies used for the algorithm.
Comment 4: On Figure 5 I see the heatmap, but I dont see, how the algorithm can distinguish the heat from human body and heat from electronical devices.
Response: We greatly appreciate the reviewer’s insightful comments and suggestions. These have significantly contributed to improving the quality of our manuscript. All comments have been carefully addressed, and the revisions are highlighted in red in the manuscript. Below is a detailed point-by-point response:
Comment 5: Please write, which thermal and normal camera was used for the test.
Response:Thank you for your valuable feedback. Section 3.6, titled Data Collection Cameras, has been added to the "Materials and Methodology" section on page 9.
Comment 6: The chapters 5-6-7. are too short, please extend it.
Response: Thank you for your valuable feedback. We have extended Chapters 5, 6, and 7 with additional discussion, limitations, and conclusions. The updates include a more in-depth analysis of the results, a comprehensive discussion of limitations (e.g., privacy concerns, challenges in low-light conditions), and expanded recommendations for future research. These changes can be found on pages 18-20.
Comment 7: Please revise it with native English speaker.
.
Response: We appreciate your feedback regarding the quality of the English language. We have revised and enhanced the writing accordingly. Additionally, all accepted papers undergo professional editing by MDPI prior to publication. From our side, we have also conducted a comprehensive review of the English language.
Finally, thanks again for your thoughtful review. We believe we have responded satisfactorily
to your concerns. Hope that the revised manuscript meets your expectations.

Reviewer 2 Report
Comments and Suggestions for Authors
Contributions:
This paper presents a computer vision-based approach designed to enhance product locating and sales strategies in physical retail stores through real-time customer behavior analysis. My comments are as follows:
- This study can be regarded as a student’s project report rather than a research paper.
- The methods are all known. The novelty is limited.
- The accuracy rate, precision rate, and recall rate for the customer number should be presented.
- The accuracy of the customer movement should be provided and analyzed.
- Why do the authors utilize YOLOv5 for object identification? Some novel versions are available.
- (Page 9) Algorithm 1 should be presented by pseudo-code.
- (Page 12) The sub-grid lines and title should be removed in Fig. 8. Figure 10 also has the same problem.
- (Page 1) Retail is not adequate as a keyword.
- (Pages 4 and 5) Table 1 should be presented on the same page for comparison.
- (Page 5) “figure 1” should be revised as “Figure 1”. Please check the usage throughout this paper.
- (Page 6) “table 1” should be revised as “Table 1”. Please check the usage throughout this paper.
- (Page 8) The presentation of the block shape for “ if high traffic?” is wrong.
The quality of the English language is not acceptable.
Author Response
Response to Reviewer 2 Comments-Round 2
Thank you for your extremely useful comments and suggestions. Based on your feedback, the paper has been thoroughly revised. We have made every effort to address all comments in the updated manuscript. The most significant changes are highlighted in yellow in the revised document. A point-by-point response to each comment is provided below:
Comment 1: This study can be regarded as a student’s project report rather than a research paper.
Response: Thank you for your feedback. To address this, we have expanded the introduction to provide a stronger scientific foundation and highlight the specific challenges faced by traditional stores in real-time customer behavior tracking compared to online retailers. This includes a more detailed analysis of the literature that emphasizes these unique challenges. See pages 1 and 2 for updates.
Comment 2: The methods are all known. The novelty is limited.
Response: We appreciate your suggestion. The literature review has been updated to incorporate recent studies from 2023–2024 on AI-driven retail monitoring systems. These additions highlight the relevance of our approach and position it within the current research landscape. Please refer to pages 4 and 5 for the updates.
Comment 3: The accuracy rate, precision rate, and recall rate for the customer number should be presented.
Response: Thank you for your suggestion. A new Section 4.8, titled Performance Metrics for Customer Detection and Tracking, has been added. This section provides detailed metrics such as accuracy, precision, and recall to evaluate the performance of customer detection and tracking. Please refer to page 17 for the updates.
Comment 4: The accuracy of the customer movement should be provided and analyzed.
Response: We have addressed your comment by including Section 4.9, Comparison with Other AI-Based and Conventional Methods. This section analyzes the accuracy of customer movement tracking and provides a comparative evaluation of our system against other models. See page 18 for detailed insights.
Comment 5: Why do the authors utilize YOLOv5 for object identification? Some novel versions are available.
Response: YOLOv5 was selected for its optimal balance between accuracy and computational efficiency, which makes it well-suited for real-time applications in retail environments. To provide further clarification, a new section, 3.5, titled Justification for Using YOLOv5, has been added on page 9.
Comment 6: (Page 9) Algorithm 1 should be presented by pseudo-code.
Response: Response: I have added a new section, 3.8, titled *Pseudocode for Customer Behavior Tracking and Heatmap Generation*, on page 10. Additionally, Algorithm 1 has been revised and is now presented in pseudocode format for better clarity. Please refer to page 10 for the updated content.
Comment 7: (Page 12) The sub-grid lines and title should be removed in Fig. 8. Figure 10 also has the same problem.
Response: Thank you for the suggestion. Figures 8 and 10 have been revised to remove sub-grid lines and adjust titles as per the reviewer’s recommendation. Please see pages 7 and 8.
Comment 8: (Page 1) Retail is not adequate as a keyword.
Response: The keyword "Retail" has been replaced with the more specific term "Heatmap Analysis" to better reflect the focus of the study. This change can be found on page 1, in the abstract.
Comment 9: (Pages 4 and 5) Table 1 should be presented on the same page for comparison.
Response: Table 1 has been reformatted to appear on a single page, ensuring readability and comparison clarity. Please refer to the updated table on page 5.
Comment 10: (Page 5) “figure 1” should be revised as “Figure 1”. Please check the usage throughout this paper.
Response: All figures and tables have been reviewed to ensure consistent capitalization and formatting according to MDPI standards.
Comment 11: (Page 6) “table 1” should be revised as “Table 1”. Please check the usage throughout this paper.
Response: We appreciate your feedback .Similar to the previous comment, all table references have been standardized.
Comment 12: (Page 8) The presentation of the block shape for “ if high traffic?” is wrong.
Response: The block shape in the flowchart has been corrected for accuracy. Please refer to the updated figure on page 8.
Comment 13: The quality of the English language is not acceptable.
Response: We appreciate your feedback regarding the quality of the English language. We have revised and enhanced the writing accordingly. Additionally, all accepted papers undergo professional editing by MDPI prior to publication. From our side, we have also conducted a comprehensive review of the English language.
Finally, thanks again for your thoughtful review. We believe we have responded satisfactorily
to your concerns. Hope that the revised manuscript meets your expectations.

Reviewer 3 Report
Comments and Suggestions for Authors
The manuscript proposed a system for retail businesses using surveillance camera video streams to analyze customers’ behavior that can further improve the customers’ experience, quality of service, advertising, etc.
The idea is straight forward and easy to understand, and the methodology is simple (looks like). While some parts are not well introduced, like heatmap updating, and source of data in the experimental section.
More specifically:
-
“where cooler colors signal low-engagement areas and 17 warmer colors represent high-traffic zones,” Line 17: There is no graphical content in the abstract. I think it doesn't make sense to mention about the color schema here. You can explain in the caption of heatmap or in main text.
-
The similarity between Fig.2 and Fig.3 are over 90%. FIg3 seems like a simplified version without annotations.
-
“return frame” Alg1. def_process_frame last line, you return the entire function when the first track is confirmed. Seems incorrect.
-
As a key of this paper, update_heatmap was not elaborated on. It was called in algorithm 1, but the function was not defined.
-
“Conversion Rate (%) Sales Figures ($)” Table 3, where did you get conversion rate and sales data? How did you collect those, has the PII been removed, and what helps to correspond the purchase transaction to each buyer's trajectory?
-
“Figure 5. Detection Results Illustrated in Heatmaps” The heatmap looks like some islands, doesn't look like a trajectory. Could the author elaborate more about it? Since the heatmap update function is not defined, I am not clear about it.
-
“Table 3. Customer Trajectory and Dwell Time Data” How to derive those data from the proposed system?
(1) human-readable path from trajectory.
(2) Engagement level.
For your future work, I think your method can be further extended to UAV surveillance camera
Line 380: There is one potential application of a similar work that you can try, using UAV data to analyze spatial heatmap of cars or pedestrians.
As a reference:
Bowman, Jordan, et al. "UAS Edge Computing of Energy Infrastructure Damage Assessment." Photogrammetric Engineering & Remote Sensing 89.2 (2023): 79-87.
There are many manner issues in formatting and language that should be corrected and paid attention to:
-
Misuse of capitalized words in the text:
for example:
Line 108, "Proposed System" please uses lower case
please check the figure / table in all text. They should be correctly capitalized.
2. typos:
“account” (Shili et al., p. 4) L154: is this `counting`?
Line 274: “costumer” -> costumer?
Line 324: Generated -> Generation
please correct rest of them, I haven’t check every word.
Author Response
Response to Reviewer 3 Comments-Round 2
Thank you for your extremely useful comments and suggestions. Based on your feedback, the paper has been thoroughly revised. We have made every effort to address all comments in the updated manuscript. The most significant changes are highlighted in yellow in the revised document. A point-by-point response to each comment is provided below:
Comment 1: “where cooler colors signal low-engagement areas and 17 warmer colors represent high-traffic zones,” Line 17: There is no graphical content in the abstract. I think it doesn't make sense to mention about the color schema here. You can explain in the caption of heatmap or in main text.
Response: Thank you for your valuable feedback. The mention of the color schema has been removed from the abstract. Instead, it is now elaborated on in the main text and the caption of Figure 5, where the heatmap is discussed in detail. Please see page 1 for the abstract and page 13 for the updated description in the figure caption.
Comment 2: The similarity between Fig.2 and Fig.3 are over 90%. FIg3 seems like a simplified version without annotations.
Response: Thank you for pointing this out. Figures 2 and 3 have been revised to avoid redundancy. Figure 3 has been annotated further to clarify its unique contribution and distinguish it from Figure 2.
Comment 3: “return frame” Alg1. def_process_frame last line, you return the entire function when the first track is confirmed. Seems incorrect.
Response: Thank you for pointing out this issue. The algorithm has been thoroughly reviewed and corrected. The revised algorithm is now detailed in Section 3.9, Algorithm for Customer Behavior Tracking and Heatmap Analysis Using YOLOv5 and DeepSORT. Please refer to pages 10–12 for the updates.
Comment 4: As a key of this paper, update_heatmap was not elaborated on. It was called in algorithm 1, but the function was not defined.
Response: We appreciate your observation. The function `update_heatmap` was indeed not elaborated upon in the initial submission. This oversight has been addressed, and the function is now defined and explained in detail as part of the updated algorithm described in Section 3.9. Please see page 11 for the details.
Comment 5: “Conversion Rate (%) Sales Figures ($)” Table 3, where did you get conversion rate and sales data? How did you collect those, has the PII been removed, and what helps to correspond the purchase transaction to each buyer's trajectory?
Response: Thank you for your inquiry. The conversion rates and sales data presented in Table 4 are simulated values designed to demonstrate the potential applications of the proposed system. To address this further, we have added Section 3.4, titled Data Collection and Privacy Considerations, which provides additional clarification on these aspects. Please refer to pages 8 and 9 for the details.
Comment 6: “Figure 5. Detection Results Illustrated in Heatmaps” The heatmap looks like some islands, doesn't look like a trajectory. Could the author elaborate more about it? Since the heatmap update function is not defined, I am not clear about it.
Response: The appearance of the heatmap represents customer density rather than individual trajectories. The methodology has been updated to elaborate on how the data is aggregated to generate the heatmap, emphasizing high-engagement zones rather than detailed paths. These details have been added to Figure 5. Please see page 13 for the updates.
Comment 7: “Table 3. Customer Trajectory and Dwell Time Data” How to derive those data from the proposed system?
(1) human-readable path from trajectory.
(2) Engagement level.
For your future work, I think your method can be further extended to UAV surveillance camera
Line 380: There is one potential application of a similar work that you can try, using UAV data to analyze spatial heatmap of cars or pedestrians.
As a reference:
Bowman, Jordan, et al. "UAS Edge Computing of Energy Infrastructure Damage Assessment." Photogrammetric Engineering & Remote Sensing 89.2 (2023): 79-87.
Response: The data in Table 3 is derived by processing customer trajectories captured by the proposed system, which utilizes YOLOv5 for object detection and DeepSORT for tracking customers in real-time. As customers move throughout the store, their paths are tracked and recorded. The system identifies key sections, such as entrances, aisles, and checkout areas, and logs the sequence of sections visited. Regarding future work, we plan to extend our method to incorporate UAV surveillance cameras for spatial heatmap analysis.Please refer to Section 7, Conclusion and Future Work, for additional insights on the potential extension of our methodology. Please see page 20.
Comment 8: Th There are many manner issues in formatting and language that should be corrected and paid attention to:
Misuse of capitalized words in the text:
for example:
Line 108, "Proposed System" please uses lower case
please check the figure / table in all text. They should be correctly capitalized.
- typos:
“account” (Shili et al., p. 4) L154: is this `counting`?
Line 274: “costumer” -> costumer?
Line 324: Generated -> Generation
please correct rest of them, I haven’t check every word.
Response: We appreciate your valuable feedback regarding the quality of the English language. The text has been thoroughly reviewed, and all necessary corrections and enhancements have been made to improve its clarity and accuracy.
Finally, thanks again for your thoughtful review. We believe we have responded satisfactorily
to your concerns. Hope that the revised manuscript meets your expectations.

Round 2
Reviewer 2 Report
Comments and Suggestions for Authors
The authors have improved the quality of this paper. I think it can be accepted for publication after minor revision.
Minor comment:
The sub-grid lines should be removed in Fig. 8.
Author Response
Response to Reviewer 2 Comments-Round 3
Thank you for your extremely useful comments and suggestions. Based on your feedback, the paper has been thoroughly revised. We have made every effort to address all comments in the updated manuscript. The most significant changes are highlighted in yellow in the revised document. A point-by-point response to each comment is provided below:
Comment 1: The authors have improved the quality of this paper. I think it can be accepted for publication after minor revision.
Minor comment:
The sub-grid lines should be removed in Fig. 8.
Response: Thank you for your valuable feedback. We have carefully addressed your comment and updated Fig. 8 to remove the sub-grid lines. The revised figure can be found on page 15 of the updated manuscript
Finally, thanks again for your thoughtful review. We believe we have responded satisfactorily
to your concerns. Hope that the revised manuscript meets your expectations.